# Exploring Precursors of Construction Accidents in China: A Grounded Theory Approach

**DOI:** 10.3390/ijerph18020410

**Published:** 2021-01-07

**Authors:** Zhen Li, Rui Mao, Qing Feng Meng, Xin Hu, Hong Xian Li

**Affiliations:** 1School of Management, Jiangsu University, 301 Xuefu Road, Zhenjiang 212013, China; janeli@ujs.edu.cn (Z.L.); 2221810029@stmail.ujs.edu.cn (R.M.); 2School of Architecture and Built Environment, Deakin University, 1 Gheringhap Street, Geelong, VIC 3220, Australia; xin.hu@deakin.edu.au (X.H.); hong.li@deakin.edu.au (H.X.L.)

**Keywords:** precursors, precursor management, construction accidents, grounded theory, China

## Abstract

The implementation of precursor management can improve safety performance of construction projects through effectively managing the correlations between construction accidents and their precursors. However, a system of comprehensive knowledge about what precursors mean within the context of construction safety is still lacking. This study aims to capture the nature of precursors in the construction industry and explore the process of a precursor event evolving into a construction accident to fill this gap. Based on 135 construction accident reports in China, this study adopts grounded theory to identify different types of accident precursors and explore their interactions with the development of the accident. An indicator system of precursors for construction accidents was developed, which included two major categories of precursors: behavioral factors and physical factors and five minor categories of precursors: individual behavior factors, organizational driving factors, objective physical factors, construction environmental factors, mechanical equipment factors. In addition, a precursor management strategy that includes the three stages of identification, response and effectiveness testing was established. The results of the study reveal the correlations between precursors and construction accidents, which can promote construction professionals’ better understanding about precursors and improve their capabilities of managing precursors in practice.

## 1. Introduction

Accidents occur frequently in the construction sector due mainly to the uniqueness, complexity and uncertainty of construction activities [1]. In Australia, the construction sector is one of the most hazardous industries with a high fatality rate, accounting for 16% of work-related fatalities in 2017 [2]. According to the statistics of the Emergency Management Department of the People’s Republic of China in the first half of 2018, there were 1732 accidents and 1752 fatalities in the construction industry [3]. The Chinese construction sector is encountering challenges regarding safe construction due to such reasons as a lack of training, poor safety awareness and ineffective safety supervision [4]. Consequently, a large number of reports about the occurrence of construction accidents can be found in China. For instance, the number of construction accidents increased from 442 to 634 during 2015–2016, resulting in 735 fatalities in 2016 [5].

Various reasons can lead to the occurrence of construction accidents, and one of the major ones is the lack of recognition about accident precursors [6]. Accident precursors are the conditions, events, and sequences that precede and lead up to accidents [7]. It has been commonly accepted that it is of great importance to effectively manage precursors in practice given that their effective management provides an accident early warning system to help construction managers take precautions in order to avoid accident occurrences [8].

Though the importance of effectively managing precursors in the construction industry, an in-depth identification and understanding of precursors is still lacking. To address the research gap, this study identified precursors of construction accidents through a content analysis of 135 representative construction accident reports in China. The relationships between precursors and the occurrence of construction accidents are analyzed, and an indicator system of precursors was developed. In addition, the management of precursors is also explored. The study promotes a better understanding about construction safety management from the perspective of precursors. It also helps construction professionals formulate effective strategies of managing precursors, which will finally promote safe construction.

## 2. Literature Review

### 2.1. Construction Accidents and Safety Management

The importance of safety management in the construction industry has attracted much research attention. For instance, Fang identified impacting factors of on-site safety management performance such as labour and training related factors [9]. Saurin retrieved and analysed building safety best practices from the perspective of cognitive systems engineering [10]. The previous related studies generally concluded that fall from heights on construction sites is one of the most important reasons [11]. Scholars also conducted explorations about reasons of resulting in construction hazards from different perspectives such as ergonomic evaluations of suspension brackets [12,13], electrical applications [14], steel structures [15] and risk-taking behavior [16]. Recently, safety climate significantly influences safety performance, making research in the field of safety climate a vital step toward raising safety levels at construction sites [17]. Furthermore, recent authors have indicated the necessity of use multi-methodological, multi-management and multi-criteria methods analysis under different perspectives [18,19], such as biomechanical risk analysis [20]. These studies have contributed to a better understanding of construction safety management and have utilized management practices to risk analysis.

### 2.2. Precursors and Their Management in Construction Projects

Precursors are “conditions, events, and sequences that lead to an accident before the accident occurs”. Precursor management has been adopted in a variety of high-risk industries such as nuclear power, aerospace, chemistry and medical insurance to predict and prevent accidents [7]. For instance, Bier and Mosleh analysed the precursors of the accident in the space shuttle O-ring that was damaged before the Challenger disaster [21]. Kyriakidis et al. explored the correlations between precursors, accident levels, injury and death levels, and safety maturity, and identified six categories of precursors including human performance, technical failure, passengers, fire, malicious acts and management practices [22].

The management of precursors has also been explored in the construction sector. For instance, Fu et al. conducted a preliminary exploration about the threat of safety hazards and laid the foundation for further research on the prediction of safety risks on the construction site [23]. Wu et al. revealed a system mechanism to interrupt and prevent precursors in construction sites. Based on the method of case-based reasoning [8], Liu et al. developed a data mining system used for identifying accident precursors in the construction industry [24]. Melo et al. explored the application of unmanned aerial systems in the safety behaviour monitoring of construction sites to enhance the recognition of precursor events [25]. Li et al. developed an active construction management system for monitoring and identifying precursors to improve safety performance [26]. Despite of these research efforts, the correlations between precursors and construction accidents are still not very clear, and there is a need of comprehensive exploration of accident precursors in the construction industry.

### 2.3. Grounded Theory in Construction Sector

Grounded theory has been widely adopted in various social research fields. The method emphasizes the combination of different analysis activities to reveal the nature of investigated issues, such as constant comparison, abstraction, conceptual thinking and analysis, categorization, theory construction, and exploring relations between data [27]. In the field of construction precursor management, grounded theory can help retrieve precursors, determine precursor categories, and construct theoretical systems by classifying and abstracting coding combinations. In addition, it can sort out the relationships between precursor categories, which helps propose effective precursor management strategies. Its suitability for exploring precursor-related issues in the construction safety management research has been confirmed [21]. For instance, Zhou et al. adopted grounded theory to explore the nature of precursors based on collapse incidents of subway engineering projects [28]. Shojaei and Haeri used grounded theory to investigate the risks of the whole life cycle of engineering projects, and proposed a comprehensive supply chain risk management approach for construction projects [29]. Wu et al. adopted grounded theory to explore owners’ roles in the construction safety management with the identification of four safety leadership categories and four safety management chains [30].

## 3. Research Method

In this study, using grounded theory includes the two stages of data collection and data analysis. Data collection refers to collecting accident reports from the Chinese construction industry, and data analysis includes the three steps of open coding, axial coding, and selective coding. Figure 1 shows the procedure of using the grounded theory method in this study. A theoretical saturation and experts test was additionally conducted to ensure that the core categories and theoretical systems are saturated.

### 3.1. Data Collection

The construction accident reports were retrieved and collected from the official website of the Ministry of Housing and Construction of the People’s Republic of China in the 2020s. This can ensure the accuracy and authority of data collected and used in this study. The keywords adopted to search for construction accident reports include “building accident report”, “collapse accident report”, and “falling accident report”. The criteria for the selected accident report must include the basic situation of the accident unit, the accident occurrence, the accident site investigation and technical analysis, the casualty of the accident, and the direct and indirect cause analysis of the accident, ensure the integrity and referability of subsequent incident analysis. Finally, 135 representative accident reports were selected and used in this study. All the used accident cases in this study occurred during 2013–2020. 

Table 1 shows the key features of these reports, including accident types, accident locations, project types, and casualties. As shown in Table 1, the majority of these accidents involve personnel falling from high altitude (24), lifting equipment collapse (31), objects falling from high altitude (26), and building collapse (29). Additionally, most of these accidents occurred in the third-tier and fourth-tier cities. Moreover, the top-ranked project types are plant construction and maintenance project (35), commercial building project (25), and civil residential project (23). It also can be found that most accidents resulted in at least one construction worker death.

### 3.2. Data Analysis

#### 3.2.1. Open Coding

Open coding is the process of decomposing and comparing the original text to conceptualize and categorize the encoding process [27]. It organizes the initial sentences with the same or similar meaning, finds out the identifiable phenomenon, and pastes the initial code or label. Then, it gathers labels to form a concept and extracts categories. The retrieved information is condensed into several categories to form a conceptual system. Line-by-line or sentence-by-sentence is the most efficient way to conduct open coding analysis [31]. This step requires identifying and describing overall constructs relevant to precursors, based on the data set of 135 incidents. Constant comparison and modification are necessary while coding. One accident report is taken as an example to display the open coding in Figure 2. This information is part of a table produced by the Ministry of Housing and Construction of the People’s Republic of China. The table is directly imported into the Nvivo 12 software (QSR International United States, Burlington, MA, USA). As a result, a total of 321 referen nodes are open coded. They are the base for implementing spindle coding and selective coding.

#### 3.2.2. Spindle Coding

In order to form a representative and conceptual code system, spindle coding was performed on the basis of open coding. Spindle coding is a set of procedures that data were put back together in new ways after open coding. The purpose of spindle coding is to aggregate the free nodes from the previous step and group them into parent nodes and child nodes [32]. Some similar nodes are merged and renamed. The first level nodes are often called parent nodes. The others are child nodes. A child node has one parent node at most. For example, the open coding notes of “violation of crossing the railing to the crane running area”, “overhauling the worker’s illegal operation”, and “simplifying the operation of the specific operation” can form the spindle paradigm of the worker’s illegal operation behavior. Thus, the contents of accident reports were conceptualized and categorized. Figure 3 and Figure 4 illustrate parent codes and the first level of child nodes.

#### 3.2.3. Selective Coding

Selective coding is to identify the core categories and connect different categories, which refers to the process of conceptualizing the untapped and complete scope. The objective of selective coding is to develop a single storyline around which everything else is draped [33]. In the study, based on the selective coding, the precursors are divided into five categories, including individual behavior factors, construction environment factors, mechanical equipment factors, organizational drivers, and objective physical factors. However, it is necessary to further explore the complexity of the multiple relationships and capture the nature of precursors. The path diagram of precursor development in construction and the management of accident precursor are put in the discussion for research.

### 3.3. Theoretical Saturation Test

To ensure the accuracy and completeness of the research results, 5 of 135 accident reports were selected and used in the theoretical saturation test. The theoretical saturation test is the process of verifying the integrity and universality of the prior analysis results [34]. In this study, this test is conducted through re-coding the remaining five construction accident reports (including open coding, spindle coding, and selecting coding) to identify accident precursors, and comparing the re-coding results with the previous analysis results. The theoretical saturation test results showed that there are no new concepts and categories other than those identified. In order to ensure the authority and representativeness of the research results, three construction industry experts (including two Chinese construction engineers and a university professor) were invited to evaluate the coding results, all of which indicated that the coding results are logical and factual.

## 4. Findings

Table 2 shows the open coding of the selected and used accident reports with the identified spindle coding results. Their frequencies were also identified and shown in it. It can found that the mostly identified spindle codes are lack of checking, worker attitudes and opportunism, incomplete formalities, and lack of equipment testing, with all these frequencies being more than 30. There are also some other identified spindle coding results, such as lack of government regulation (28) and regulatory loopholes (28).

Table 3 shows the indicator system of the precursor influencing factors for construction projects. By counting the frequency of various precursor events occurrence in the open coding process, it can be determined the proportions of individual behavioral factors (36.2%), organizational drivers (29.2%), objective physical factors (15.6%), construction environmental factors (8%) and mechanical equipment factors (10.8%) in the indicator system. As shown in Table 3, the factors can be grouped into the two categories of behavioral factors and physical factors, and the indicator system is dominated by behavioral factors and supplemented by physical factors. The two categories can be further grouped into several sub-categories which can be explained by open coding results. It also shows that Personal safety awareness is the most important impacting factor of accident precursors in the construction industry.

By counting the frequency of various precursors occurrence in each type of accident, it can be found that different types of accidents have different leading precursors. As shown in Table 4, the collapse and high drop accidents accounted for the highest proportion of construction accidents, 44.4% and 37% separately. Lack of safety supervision and weak awareness of safety are identified as the most important precursors that result in collapse accidents, 29 and 32 times separately, whereas imperfect implementation of safety management system is most important for falling from a height accidents. Weak awareness of safety and lack of inspection of infrastructure are identified as two key precursors for fire accidents, 16 and 15 times separately. Compared with electric shock accidents, Failure to follow the prescribed operating procedures is the most important. 

Based on the different strategies of dealing with accidents and severity of accidents, five typical accidents (numbered P1–P5) were identified and retrieved from the 135 accidents to analyze their reasons (Table 5). As shown in Table 4, the influencing degree of accidents is impacted by precursor discovery, accident responses and strategies, and the suitability of these responses and strategies. Therefore, the management of precursors can be divided into the three stages of identification, response, and effectiveness testing (Figure 5):

*Identification*: If a precursor is not enough to cause the accident and the relevant subjects have not been identified in time, this will not lead to a serious accident. However, if a precursor is the main factor that directly causes the accident and is not recognized, the precursor will finally result in accidents with serious consequences. Such as: P1 precursor is not discovered, the precursor continues to develop into a critical event, but due to insufficient necessary conditions to cause the accident, it does not constitute a serious accident, there is opportunism. P2 is also undiscovered and contains the precursors that mainly caused the accident. The precursors developed, serious accidents occured. Therefore, the identification stage is the beginning of the entire development process of precursors.

*Response*: When a precursor is identified, whether workers take action against it or not will directly impact the development and evolution of the precursor. If a precursor is the main cause of accidents and workers do not take any action to deal with it, the precursor event itself will directly lead to accidents with serious consequences. Such as P3: The workers have found that the construction lifts are not equipped with safety gates and are in an unsafe state. However, no measures have been taken to prevent the precursor, and the precursor can cause an accident, leading to a serious accident. The reaction stage is the second stage of the development process of precursor.

*Effectiveness testing*: The effectiveness of the action used to deal with precursors will impact the performance of precursor management. The construction process is affected by a variety of factors that are related to environment, construction equipment, and construction atmosphere. In addition, workers’ ways of responding to precursors are different. It becomes imperative to test the usefulness of adopted action. If precursors are identified and reasonable and effective risk response measures are taken, construction accidents can be largely avoided. Such as P4 and P5: the worker only reported to the construction team leader and has not been directly reported to the effective influencer, and the adoption of invalid evasive measures has led to serious construction consequences. No effective measures are generated, leading to serious accidents. P5: Worker A found precursor and stopped the construction work B, reported to the superior supervision department, the logistics department in time. There were no serious consequences. Effectiveness testing is the last link in the development of precursors.

## 5. Discussion

By reviewing of 135 construction accident reports, it indicates that the performance of individual behavioral factors is affected by organizational-driven precursors, and organizational drivers internalize individual behaviors. Such as P2 and P4 in Table 5, the indirect cause of accidents is related to the organization, whereas the direct cause is usually individuals’ poor performance. Before the occurrence of construction accidents, individual behavioral precursors emerged. However, individual behaviors were subjective and variability, and impacted by organizational drivers and conditions. 

The physical factors of precursors include objective physics, construction environment, and mechanical equipment factors. P1 and P2 and P5 accident reports show that the objective conditions of construction will directly lead to the occurrence of engineering accidents, which is the direct cause of accidents. In addition, the physical precursors can be avoided or reduced by behavioral precursors. In the case of physical precursors as the dominant factor, the physical precursors can directly lead to construction accidents, but can be identified and avoided through individual behavior and organizational drivers. It is important to implement real-time identification of pre-cursors during the construction process to facilitate the management of precursor events. Based on the result and analysis of typical cases, it is found that in the order of organizational-driven precursor, individual behavior precursor, and objective physical precursor, the detectability and recognizability continue to increase, but the internal influence and ambiguity continue to weaken like Figure 6. 

Different from discussing the relationships and differentiations accident, near miss, and unsafe behavior/condition [25], this study analysis of different types of building accidents regarding major precursors. It found that collapses and falls from high places are two most frequently occurred accident types, accounting for 44.4% and 37% respectively from Table 4. These two types of accidents have the characteristics of high incidence, changeable occurrence, difficult to avoid. Therefore, it is important to further analyze the two types of accidents to have a better understanding about them. 

Individual behavioral and organization-driven factors are the dominant precursors in the high-altitude falling accidents from Figure 7, accounting for 30% and 33%, respectively. The reasons to these accidents include weak safety awareness, lack of safety handling abilities, and some subjective factors that affect behaviors. This kind of precursor is not easy to identify before the occurrence of accidents, and lacks methods of real-time monitoring. In order to avoid accidents caused by such precursory events, one of the suggested strategies is safety training which can improve workers’ safety consciousness, strengthen safety supervision, and form a safety culture and atmosphere.

According to the result of the findings and Figure 5, this study considers precursor management measures from the perspective of the process of a precursor evolving into a construction accident. Compared with the traditional precursor management [8], it was divided into three stage: identification, response, and effectiveness stage.

At the identification stage, it is critical to identify precursors to take preventive measures. First, the current precursor identification methods and techniques are insufficient, which negatively impacts workers’ safety recognition in construction activities. Also, construction workers’ insufficient safety knowledge and awareness makes them un-sensitive to precursors. Thus, it is necessary to improve safety awareness and knowledge of workers, and develop effective methods and techniques for identifying precursors.

At the response stage, the response to precursors is affected by such factors as safety awareness, organizational safety atmosphere, subjective attitude, and regulatory pressure. It is suggested to organize and carry out safety responsibility training to improve construction professionals’ sense of responsibility to handle precursors. In addition, the inspection system and the supervision system should be established, and the regulations regarding the reward and punishment in the process of handling precursors should be well developed.

At the effectiveness testing stage, it is important to ensure that effective responses are proposed given that using effective response. However, a lot of uncontrollable factors impact the effectiveness of the strategies used such as the nature of precursors and the abilities of managers. It is important that all project stakeholders can work together to understand these factors and handle precursors in timely, effective and reasonable ways.

## 6. Limitations

The study has some limitations that should be well considered and improved in future studies. For instance, the study is based on the grounded theory which is more based on qualitative analysis. Future studies can consider the usefulness of quantitative analysis methods to explore the research topic from a different perspective. More efficient and real-time identification technology can also be further explored. Combining with the prevention and management of accident precursors to establish a complete set of behavioral and physical-based early warning mechanisms is a direction that can be explored. Despite so, the results of this study can promote a better understanding about precursor management, which will finally facilitate the safety management in the construction industry.

## 7. Conclusions

Construction accidents occur frequently due to the dynamic and temporary characteristics of construction projects. It is great importance to select and adopt effective strategies to prevent them. Precursor management is an effective way of predicting risks and avoiding accidents. Based on the collected 135 accident reports from the Chinese construction sector, this paper uses grounded theory to forms a precursor indicator system, and the core elements of this system which included two major categories of precursors: behavioral factors and physical factors and five minor categories of precursors: individual behavior factors, organizational driving factors, objective physical factors, construction environmental factors, mechanical equipment factors. Based on analyzing the process of a precursor evolving into a construction accident, precursor management strategy that includes the three steps of identification, response, and effectiveness testing is proposed to facilitate the safety management on construction sites. To effectively deal with precursors, strategies should be well designed at these different stages.

In addition, based on the analysis of typical cases, it is found that in the order of organizational-driven precursor, individual behavior precursor, and objective physical precursor, the detectability and recognizability continue to increase, but the internal influence and ambiguity continue to weaken. Among the two major types of accidents, collapse and falls, individual behavior and organizational drive account for the highest proportion. Therefore, during the security management of precursor, while continuously deepening the organization’s security atmosphere and personal safety awareness, advanced precursor identification technology is used to identify objective physical precursor. The results of the study reveal the correlations between precursors and construction accidents, which can promote construction professionals’ better understanding about precursors and improve their capabilities of managing precursors in practice.

## Figures and Tables

**Figure 1 ijerph-18-00410-f001:**
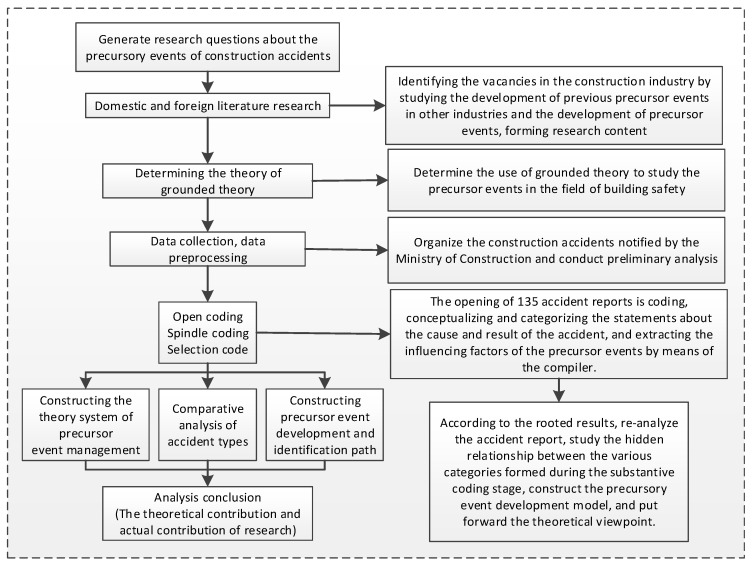
The procedure of using grounded theory in this study.

**Figure 2 ijerph-18-00410-f002:**
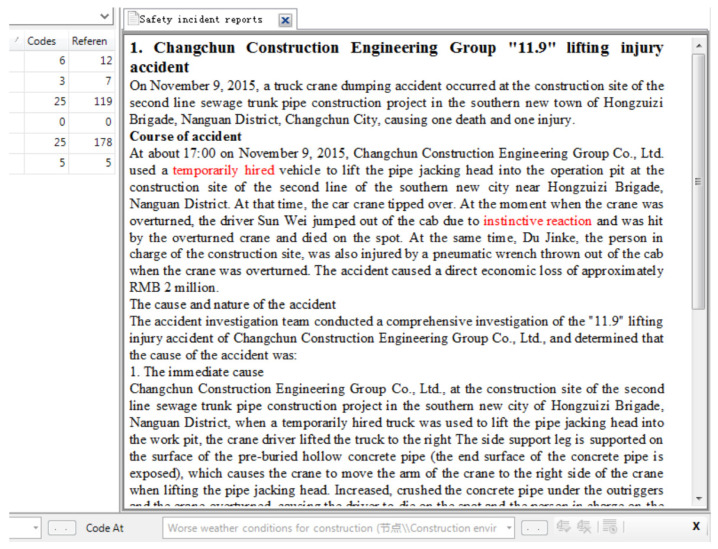
A part of a safety accident report and its analysis process.

**Figure 3 ijerph-18-00410-f003:**
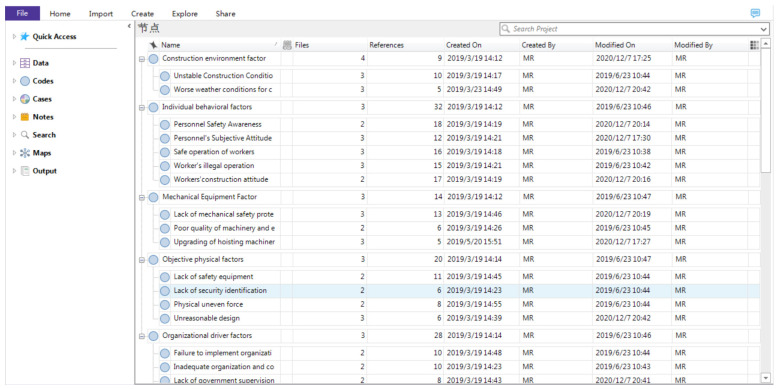
Selective coding results by using Nvivo12 (Part 1). The Chinese in the picture means node.

**Figure 4 ijerph-18-00410-f004:**
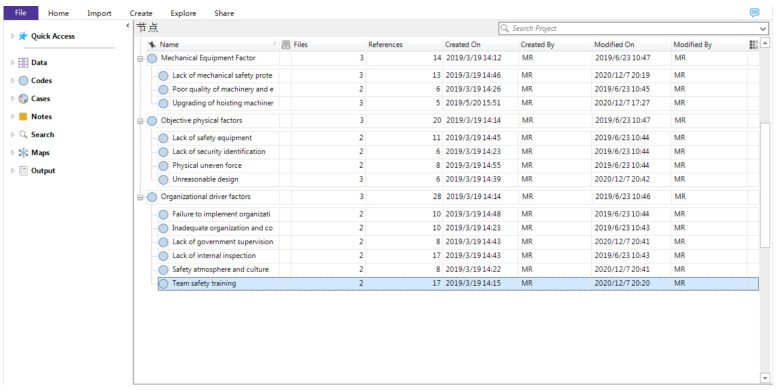
Selective coding results by using Nvivo12 (Part 2). The Chinese in the picture means node.

**Figure 5 ijerph-18-00410-f005:**
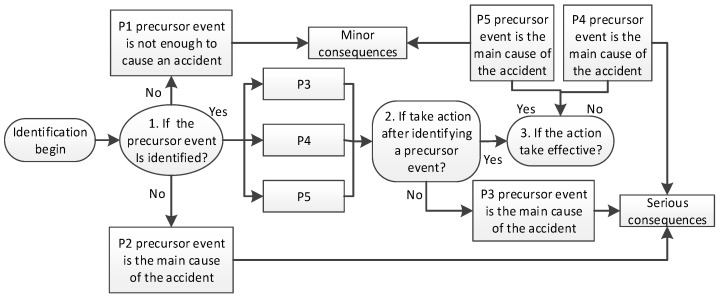
The roadmap for the management of precursors.

**Figure 6 ijerph-18-00410-f006:**
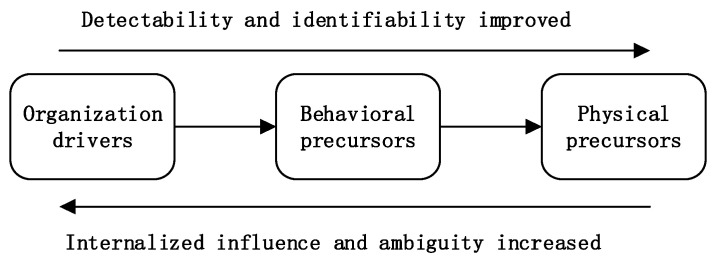
Diagram of changes in the three types of precursors.

**Figure 7 ijerph-18-00410-f007:**
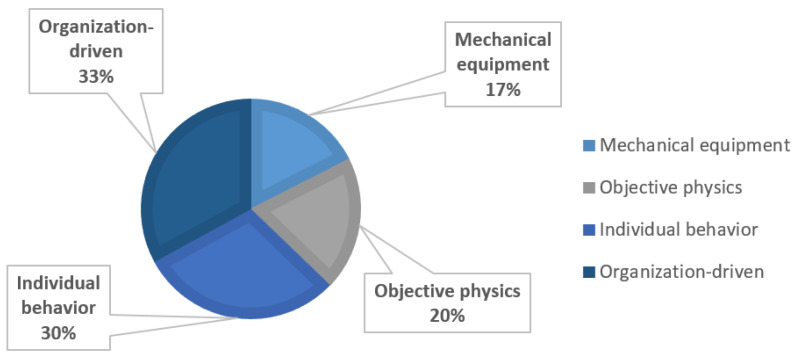
Types of precursors in the high-altitude falling accidents.

**Table 1 ijerph-18-00410-t001:** Statistical information on collected construction accidents.

**Accident Type**	**Frequency**	**Construction Project Type**	**Frequency**
Lifting equipment collapse	31	Urban infrastructure project	24
Building collapse	29	Plant construction and maintenance project	35
Object falling from high altitude	26	Civil residential project	23
Personnel falling from high altitude	24	National Engineering Construction Project	9
Fire accident	14	Public civil construction project	19
Electric shock accident	11	Commercial building project	25
**Statistics of Accident Location**	**Frequency**	**Statistics of Casualties in Construction Accidents**	**Frequency**
First-tier cities	21	Minor wound	21
Second-tier cities	31	Serious injury	16
Third-class cities	49	One to two deaths	41
Fourth-tier cities	34	Three to five deaths	32
		More than five men died	25

**Table 2 ijerph-18-00410-t002:** The open coding with the identified spindle coding results.

Code	Original Statements Retrieved from Accident Reports	Frequency	Spindle Coding
H13	The supervisor has not reviewed the authenticity of the data, the unqualified construction team’s operating qualifications and the operation permit.	33	Lack of checking
H29	Incomplete construction procedures exist in construction units	32	Incomplete formalities
H27	The inspectors have a fluke mentality and an incorrect attitude towards the inspection of safety links.	30	Worker attitudes and opportunism
H20	Lack of real-time testing of construction workers to implement unsafe behavior instruments	30	Lack of equipment testing
H14	The government’s safety management department has violated regulations and violated its regulatory obligations.	28	Lack of government regulation
H28	The supervision unit failed to stop the structural design and checking calculation of the actual operation platform of the project in time.	28	Regulatory loopholes
H22	The safety production rules and regulations are not perfect, and the safety production responsibility system is not in place.	28	Organizational system is not implemented
H11	The old crane is out of repair and has not passed the safety check.	27	The quality of the lifting machinery is not good
H6	Workers will not actively report the precursor of the accident	25	Human attitude
H21	The crane does not have regular maintenance and inspection	24	Equipment maintenance
H25	Weather conditions lead to poor construction conditions and mechanical instability.	23	Construction weather environment
H26	The tower operator did not wear a seat belt.	23	Missing safety equipment
H4	Managers and practitioners have a weak security awareness	23	Safety consciousness
H24	Responsible persons and managers do not have the safety production knowledge and management ability of their production and operation activities.	19	Insufficient personnel capacity
H8	Not communicating effectively with the driver of the crane	19	Organizational communication
H10	Coordination work of various working groups is not in place, there are security risks	17	Coordination is not in place
H23	Scaffolding steel pipe joints are loose, no necessary support	16	Infrastructure is at risk
H15	Construction workers lack necessary safety equipment	16	Lack of safety equipment
H33	The builders lack safety awareness and throw construction waste downstairs.	15	Safety consciousness Illegal operation
H5	Temporary hiring personnel (including car hoists) fail to carry out safety technical disclosure and safety education training as required	15	Not operating according to the program Safety Training
H30	Contractor did not adjust in time according to load test standard	14	Unreal-time adjustment of equipment parameters
H34	Sequential operation failure of crane results in excessive force exerted on a crane arm.	14	Operational Sequence Error
H1	Temporary employment of crane drivers without professional training	13	Safety Training
H18	The construction materials are stacked against the wall, causing the wall to collapse due to uneven force.	9	Uneven objective physical force
H37	The safety management system is not strictly enforced and no one is in charge of the construction site.	8	Security system implementation
H9	Prohibition of missing warning signs such as crossings and danger sources	6	Missing identity
H32	Over-excavation depth of excavator leads to overturning due to uneven stress on support surface	5	Objective physical factors
H35	Construction personnel use lifts without safety devices	5	Lack of safety equipment
H3	Failure to comply with the supervision requirements, the implementation of the inspection and safety inspection system for the car cranes entering the market	5	Not operating as required
H7	Violation of the rules across the railing to the crane operating area	4	Violation operation
H17	There are no relevant regulations for maintenance procedures.	4	Lack of organizational regulations
H36	Heavy rain leads to wet and slippery ground, weakening the support strength of crane bracket, leading to crane collapse.	3	Bad weather
H2	The side arm of the crane is fixed on the unstable ground (the exposed hollow pipe on the surface is not strong enough)	3	Unstable construction conditions No safe operation
H12	The design of the support structure is unreasonable, and the support weight is not reached.	3	Unreasonable design
H16	Overhauling workers in violation of regulations will simplify the process of requiring dedicated operations	3	Violation operation
H31	1.5 cm cracks appeared in the construction floor, which did not stop construction in time for repair.	2	Illegal operation Lack of supervision
H19	The safety awareness is weak, and the crane that operates the torque limiter and the level fails is lifted in an uneven field.	1	safety consciousness

**Table 3 ijerph-18-00410-t003:** The indicator system of precursors for construction projects.

Classification	Sub-Category	The Connotation of Open Coding	Frequency
Behavioral factors 65.4%	Individual behavioral factors 36.2%	Personnel safety awareness	31
		Safe operation of workers	29
		Workers’ construction attitude	25
		Worker’s illegal operation	23
		Personnel’s subjective attitude	22
	Organizational driver factors 29.2%	Team safety training	24
		Lack of internal inspection	22
		Inadequate organization and coordination	20
		Lack of government supervision	18
		Safety atmosphere and culture	12
		Failure to implement organizational system	9
Physical factors 34.4%	Objective physical factors 15.6%	Lack of safety equipment	19
		Physical uneven force	17
		Lack of security identification	11
		Unreasonable design	9
	Construction environmental factors 8%	Unstable construction conditions	18
		Worse weather conditions for construction	11
	Mechanical Equipment factors 10.8%	Lack of mechanical safety protection	18
		Poor quality of machinery and equipment	12
		Upgrading of hoisting machinery lags behind	9

**Table 4 ijerph-18-00410-t004:** The comparison of precursors of different accident types.

Accident Types	Main Axis Representation of Leading Precursors	Frequency	Category of Precursors
Collapse—44.4%	Instability of hoisting equipment during construction	17	Mechanical equipment
	Unconscious objective physical factors	19	Objective physics
	Failure to follow the prescribed operating procedures	28	Individual behavior
	Weak awareness of safety	32	
	Safety training and qualification examination are not in place	21	Organization-driven
	Lack of safety supervision	29	
Falling from a height—37%	Lack of Safety Protection Equipment and Label	19	Objective physics
	Weak awareness of safety	25	Individual behavior
	Personal unsafe behavior of constructors	27	
	Failure to follow the prescribed procedure	30	
	Insufficient safety supervision	28	Organization-driven
	Imperfect implementation of safety management system	31	
Fire—10.3%	Building materials are not stored in accordance with regulations	11	Objective physics
	Environmental factors at the construction site	12	
	Weak awareness of safety	16	Individual behavior
	Lack of inspection of infrastructure	15	Organization-driven
Electric shock—8.1%	Lack of equipment testing	19	Mechanical equipment
	Lack of Safety Protection Equipment and Label	15	Objective physics
	Failure to follow the prescribed operating procedures	23	Individual behavior

**Table 5 ijerph-18-00410-t005:** Five typical construction accidents.

Number	Accident Details
P1	High-altitude accident: The building scaffolding is not fixed, but because the location is not densely populated, the precursor events continue to evolve and do not cause serious safety incidents
P2	Hubei Good Mansion lifting Collapse accident: the relationship workers did not find the overload of the lifting object, the problem of the lifting arm, the means of identifying the precursor, and the continued operation of the precursor and the emergence of the precursor, eventually leading to the overweight bearing of the boom, the crane collapsed, and this is a serious construction accident
P3	Chang’an Town high altitude fall accident: The workers have found that the construction lifts are not equipped with safety gates and are in an unsafe state. However, due to the tight schedule, the organization is not strict with the construction process and there is opportunism that the probability of accidents is small and there is no unsafe condition for the machinery. Disposal leads to the occurrence of a high fall accident
P4	When the worker finds that the crane driver is in the unlicensed driving state, the worker is under the pressure of the construction captain, the actual situation of the driver who lacks the driving of the license, and the atmosphere of the entire construction team not paying attention to the safe operation, the worker It is only reported to the construction team leader and informed the driver to pay attention to safety, and has not been directly reported to the owner, the contracting company, the supervisor, and the adoption of invalid evasive measures has led to serious construction consequences
P5	Worker A found that the connection of the safety rope of the high-altitude operator B was in excessive wear state, then stopped the construction work B and reported to the superior supervision department and the logistics department, and timely replaced the lifting rope and handed it to the supervision department. The matter was accountable and found that it was B’s own reason that the rope was not replaced in time, and the corresponding punishment decision was made for B

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
