# Peer review of "Exploring Precursors of Construction Accidents in China: A Grounded Theory Approach"

_ijerph, 2021, doi:10.3390/ijerph18020410_

Round 1

Reviewer 1 Report

This article presents a grounded theory to identify different types of accident precursors and explore their interactions with different accidents. The work is in the scope of the journal, however, redaction and structure should be improved as indicated below, especially the methods should be clearer; the author is recommended to identify and practice sophisticated objectives for a journal publication. The author must justify the following points:

Comment 1: the paper should be revised to highlight novelties. Please consider that this lack of novelty starts with the Abstract, Introduction, and Conclusion. The aim of this work should be clarified more clearly. What is the importance and the scientific contribution of this work?

Comment 2: Figure 2 needs to be built up within better resolution

Comment 3: Section 3 is not outlined with necessary vigor. The author needs to include sufficient methodological details in the paper and elaborate on the produced results from the proposed methods. The paper should provide enough information to be replicable by other researchers. The most important issue to be justified herein in that the author must present and explain the applied database used in Nvivo 12 to sort out the outputs? Based on Section3, it was very confused to identify the results presented in Section 4, basically Table1.

Comment 4: The conclusion section must be elaborated based on a deep analysis of the collected results and according to the identified novelty and aims of the study. More Tables and Figures are suggested to be utilized at this level of the analysis.

Comment 5: The Conclusion section is very concise and missing lots of necessary details. For example, the author needs to highlight the novelty and methods used in this work, and point out the collected results. Then, present a summary of the limitations of this research as well as the recommendation for future works. 

Author Response

Response to Reviewer 1 Comments

Point 1: The paper should be revised to highlight novelties. Please consider that this lack of novelty starts with the Abstract, Introduction, and Conclusion. The aim of this work should be clarified more clearly. What is the importance and the scientific contribution of this work?

Response 1: It is our great honor to receive this valuable comment, which is very helpful for us to better refine our novelties and scientific contribution of this work. According to your suggestion, we have re-summarized the innovations and contributions of this paper, and put them in the Abstract, Introduction, and Conclusion so that it can be more concise and clearer, as shown in the following parts.

This study aims to capture the nature of precursors in the construction industry and explore the process of a precursor event evolving into a construction accident to form a system of comprehensive knowledge about what precursors mean within the context of construction safety.

The novelty and the scientific contribution of this work is: Firstly, an indicator system of precursors for construction accidents was developed, which included two major categories of precursors: behavioral factors and physical factors and five minor categories of precursors: individual behavioral factors, organizational driving factors, objective physical factors, construction environmental factors, mechanical equipment factors. Secondly, a precursor management strategy that includes the three stages of identification, response and effectiveness testing was established. Finally, based on the analysis of typical cases, it is found that in the order of organizational-driven precursor, individual behavior precursor, and objective physical precursor, the detectability and recognizability continue to increase, but the internal influence and ambiguity continue to weaken. Among the two major types of accidents, collapse and fall, individual behavior and organizational drive account for the highest proportion. The results of the study reveal the correlations between precursors and construction accidents, which can promote construction professionals’ better understanding about precursors and improve their capabilities of managing precursors in practice.

Point 2: Figure 2 needs to be built up within better resolution.

Response 2: We want to express our gratitude to the reviewer for your meticulous and rigorous work. We have found that Figure 2 has the problem of bed resolution. In order to make the accident report data more clearer to readers, we have replaced the Figure in the form of Table.

Accident type

Frequency

Construction project type

Frequency

Lifting equipment collapse

31

Urban infrastructure project

24

Building collapse

29

Plant construction and maintenance project

35

Object falling at high altitude

26

Civil residential project

23

Personnel falling from high altitude

24

National Engineering Construction Project

9

Fire accident

14

Public civil construction project

19

Electric shock accident

11

Commercial building project

25

Statistics of Accident Location

Frequency

Statistics of casualties in construction accidents

Frequency

First-tier cities

21

Minor wound

21

Second-tier cities

31

Serious injury

16

Third-class cities

49

One to two deaths

41

Fourth-tier cities

34

Three to five deaths

32

More than five man died

25

Point 3: Section 3 is not outlined with necessary vigor. The author needs to include sufficient methodological details in the paper and elaborate on the produced results from the proposed methods. The paper should provide enough information to be replicable by other researchers. The most important issue to be justified herein in that the author must present and explain the applied database used in Nvivo 12 to sort out the outputs? Based on Section3, it was very confused to identify the results presented in Section 4, basically Table1.

Response 3: We really appreciate the reviewers for this comment. Based on your suggestions, we have made in-depth consideration and modification, adding sufficient method details:

“The criteria for the selected accident report must include the basic situation of the accident unit, the accident occurrence, the accident site investigation and technical analysis, the casualty of the accident, and the direct and indirect cause analysis of the accident.”

“It organizes the initial sentences with the same or similar meaning, finds out the identifiable phenomenon, and pastes the initial code or label. Then, it gathers labels to form a concept and extracts categories. The retrieved information is condensed into several categories to form a conceptual system. Line-byline or sentence-by-sentence is the most efficient way to conduct open coding analysis.”

“Spindle coding as a set of procedures whereby data were put back together in new ways after open coding, by making connections between constructs. The purpose of spindle coding is to aggregate the free nodes from the previous step and group them into parent nodes and child nodes. Some similar nodes are merged and renamed Some similar nodes are merged and renamed. The first level nodes are often called parent nodes. The others are child nodes. A child node has one parent node at most. For example, the open coding notes of “violation of crossing the railing to the crane running area”, “overhauling the worker’s illegal operation” can form the spindle paradigm of the worker's illegal operation behavior.”

“Selective coding is the process of selecting the core constructs. Parent nodes and child nodes from axial coding are merely descriptions of the data which cannot exhibit the multiple relationships among them. It is necessary to further explore the complexity of the multiple relationships and capture the nature of precursors. The objective of selective coding is to develop a single storyline around which everything else is draped.”

The applied database used in Nvivo 12 was analyzed by Open coding, Spindle coding, Selective coding three sequential steps.

Point 4: The conclusion section must be elaborated based on a deep analysis of the collected results and according to the identified novelty and aims of the study. More Tables and Figures are suggested to be utilized at this level of the analysis.

Response 4: It is our great honor to receive this valuable comment, which is very helpful for us to better refine our conclusion. We have applied more data analysis Figures to put forward opinions, and put more conclusions of the article in combination with typical accident reports and the results already produced.

Such as Fig. 7. Combined with analysis of typical accident cases, individual behavioral and Organization-driven factors are the dominant precursors in the high-altitude falling accidents from Fig. 7, occupying 30% and 33%. The reasons to these accidents include weak safety awareness, lack of safety handling abilities, and some subjective factors that affect behaviors.

In addition, based on the analysis of typical cases, it is found that in the order of organizational-driven precursor, individual behavior precursor, and objective physical precursor, the detectability and recognizability continue to increase, but the internal influence and ambiguity continue to weaken as like Fig. 6.

  At the same time, combining the same type of research and the previous research of Fig. 5, we proposed a precursor management strategy that includes the three stages of identification, response and effectiveness testing.

At the Identification stage, it is critical to identify precursors to take preventive measures. First, the current precursor identification methods and techniques are insufficient, which negatively impacts workers’ safety recognition in construction activities. Also, construction workers’ insufficient safety knowledge and awareness makes them un-sensitive to precursors. Thus, it is necessary to improve safety awareness and knowledge of workers, and develop effective methods and techniques for identifying precursors.

At the Response stage, the response to precursors is affected by such factors as safety awareness, organizational safety atmosphere, subjective attitude, and regulatory pressure. It is suggested to organize and carry out safety responsibility training to improve construction professionals’ sense of responsibility to handle precursors. In addition, the inspection system and the supervision system should be established, and the regulations regarding the reward and punishment in the process of handling precursors should be well developed.

At the Effectiveness testing stage, it is important to ensure that effective responses are proposed given that using effective response. However, a lot of uncontrollable factors impact the effectiveness of the strategies used such as the nature of precursors and the abilities of managers.

Point 5: The Conclusion section is very concise and missing lots of necessary details. For example, the author needs to highlight the novelty and methods used in this work, and point out the collected results. Then, present a summary of the limitations of this research as well as the recommendation for future works. 

Response 5: We really appreciate the reviewers for this comment. According to your suggestion, we have highlighted the novelty and point out the collected results. Such as “Based on the collected 135 accident reports from the Chinese construction sector, this paper uses grounded theory to forms a precursor indicator system, and the core elements of this system which included two major categories of precursors: behavioral factors and physical factors and five minor categories of precursors: individual behavior factors, organizational driving factors, objective physical factors, construction environmental factors, mechanical equipment factors.” And “Based on analyzing the process of a precursor evolving into a construction accident, precursor management strategy that includes the three steps of identification, response, and effectiveness testing is proposed to facilitate the safety management on construction sites.”

  Among the two major types of accidents, collapse and fall, individual behavior and organizational drive account for the highest proportion. Therefore, during the security management of precursor, while continuously deepening the organization's security atmosphere and personal safety awareness, advanced precursor identification technology is used to identify objective physical precursor. The results of the study reveal the correlations between precursors and construction accidents, which can promote construction professionals’ better understanding about precursors and improve their capabilities of managing precursors in practice.

  According to your suggestion about the limitations and future, we have revised a summary of the limitations of this research as well as the recommendation for future works. Such as “The study has some limitations that should be well considered and improved in future studies. For instance, the study is based on the grounded theory which is more based on qualitative analysis. Future studies can consider the usefulness of quantitative analysis methods to explore the research topic from a different perspective. More efficient and real-time identification technology can also be further explored. Combining with the prevention and management of accident precursors to establish a complete set of behavioral and physical-based early warning mechanisms is a direction that can be explored. Despite so, the results of this study can promote a better understanding about precursor management, which will finally facilitate the safety management in the construction industry.”

In all, we found the referee’s comments are quite helpful, and we revised our paper point-by-point. We thank you overall for your valuable suggestions that have helped us to craft a stronger manuscript. Thank you for your help.

Yours sincerely,

Qingfeng Meng

Name: Qingfeng Meng

E-mail: mqf@ujs.edu.cn

cc.

Zhen Li; Rui Mao; Xin Hu; Hongxian Li

Reviewer 2 Report

The paper "Exploring Precursors of Construction Accidents in China: A Grounded Theory Approach" comprise an analysis of accident precursors types and explore interactions with accidents.

Abstract

The main objectives of the study and what analysis was carried out is unclear. The type of analysis was not specified and no indication of sample size was presented. It needs to be re-worked. The results are unclear at present. The last paragraph of the abstract is not consistent as conclusion. It should be reformulated.

Introduction and literature review

The references should be actualized with current data (2020 or, in defect 2019). Literature review do not explained the root cause analysis what is usually used in construction sector.

Methods

I recommend presenting Figure 2 as table. Data should be compared to larger sampled.

Thus, general statistical indices such as the frequency, severity and incidence index, among others, could be included.

Sample considered in the article (78 accidents) it is not a representative quantity. Still, the justification for the selection has not been discussed either.

Nvio is the software applied to qualitative analysis, but qualitative analysis is not explained. The process defined in Figure 3, 4 and 5 are not necessary. The article consumes much of the method in explaining the Nvio process step by step, which is not necessary if the general qualitative techniques are explained, in addition to the coding in a table. The selection of codes is an aspect that must be verified by experts, such as an interview or questionnaire of experts, for example.

Results

Table 1, 2 and 3 should be simplified: main data should be concentrated. Typical construction accidents should comprise a detailed qualitative analysis that allows demonstrating the sufficiency of data to justify the diagram proposed in Figure 6.

Discussion

The discussion does not include limitations, relationships with previous articles and research on the subject, the techniques usually applied in construction and future proposals. An intrinsic review of this part is necessary to ensure that it is not confused with the results section.

Conclusion

The conclusion does not provide an interesting outcome. This would be possible if a thorough discussion of the current subject in construction techniques had been conducted.

Bibliography

The references do not comply with the standard of IJERPH.

Author Response

Response to Reviewer 2 Comments

Point 1: Abstract

The main objectives of the study and what analysis was carried out is unclear. The type of analysis was not specified and no indication of sample size was presented. It needs to be re-worked. The results are unclear at present. The last paragraph of the abstract is not consistent as conclusion. It should be reformulated.

Response 1: It is my great honor to get your suggestions, which is of great importance to improve the readability and logic of the Abstract. We have made the following rectifications in Abstract based on your suggestions:

First, the main objective of the study is to capture the nature of precursors in the construction industry and explore the process of a precursor event evolving into a construction accident to set up a system of comprehensive knowledge about what precursors mean within the context of construction safety.

Secondly, the type of analysis is using grounded theory which includes Open coding, Spindle coding, Selective coding. The sample size is 135 construction accident reports in China which collected from the official website of the Ministry of Housing and Construction of the People’s Republic of China.

Finally, we have reformulated conclusion to make it be consistent as last paragraph of the abstract. Correct as follows: “Construction accidents occur frequently due to the dynamic and temporary characteristics of construction projects. It is great importance to select and adopt effective strategies to prevent them. Precursor management is an effective way of predicting risks and avoiding accidents. Based on the collected 135 accident reports from the Chinese construction sector, this paper uses grounded theory to forms a precursor indicator system, and the core elements of this system which included two major categories of precursors: behavioral factors and physical factors and five minor categories of precursors: individual behavioral factors, organizational driving factors, objective physical factors, construction environmental factors, mechanical equipment factors. Based on analyzing the process of a precursor evolving into a construction accident, precursor management strategy that includes the three steps of identification, response, and effectiveness testing is proposed to facilitate the safety management on construction sites. To effectively deal with precursors, strategies should be well designed at these different stages. The results of the study reveal the correlations between precursors and construction accidents, which can promote construction professionals’ better understanding about precursors and improve their capabilities of managing precursors in practice.”

Point 2: Introduction and literature review

The references should be actualized with current data (2020 or, in defect 2019). Literature review do not explain the root cause analysis what is usually used in construction sector.

Response 2: Thanks for this comment and the consideration of these problems is very helpful to improve the rational and timely ability of this paper.

  We have updated the current data and journals in introduction and literature review, including 2019 and 2020. The literature review was revised based on the latest research. Such as: “Recently, safety climate significantly influences safety performance, making research in the field of safety climate a vital step toward raising safety levels at construction sites.” from“Makki, A. A., & Mosly, I. (2020). Determinants for Safety Climate Evaluation of Construction Industry Sites in Saudi Arabia. International Journal of Environmental Research and Public Health, 17(21), 8225.”

The research and application results of grounded theory are added in the literature review section. Grounded theory has been applied in the field of construction, and has achieved certain positive effects. It has a good effect in analyzing typical cases and analyzing research in the entire field. Such as: “ Its suitability for exploring precursor-related issues in the construction safety management research has been confirmed [18]. For instance, Zhou et al adopted grounded theory to explore the nature of precursors based on collapse incidents of subway engineering projects [25]. Shojaei and Haeri used grounded theory to investigate the risks of the whole life cycle of engineering projects, and proposed a comprehensive supply chain risk management approach for construction projects [26]. Wu et al adopted grounded theory to explore owners’ roles in the construction safety management with the identification of four safety leadership categories and four safety management chains [27].”

Point 3: Methods

I recommend presenting Figure 2 as table. Data should be compared to larger sampled.

Thus, general statistical indices such as the frequency, severity and incidence index, among others, could be included.

Sample considered in the article (78 accidents) it is not a representative quantity. Still, the justification for the selection has not been discussed either.

Nvio is the software applied to qualitative analysis, but qualitative analysis is not explained. The process defined in Figure 3, 4 and 5 are not necessary. The article consumes much of the method in explaining the Nvio process step by step, which is not necessary if the general qualitative techniques are explained, in addition to the coding in a table. The selection of codes is an aspect that must be verified by experts, such as an interview or questionnaire of experts, for example.

Response 3: It is my great honor to get your suggestions, which is of great importance to improve the readability and logic of the Methods section. Based on your suggestions, we have made the following revises to this section:

Firstly, we have presented Figure 2 as table 1 which included the frequency, severity and incidence index of accidents to make readers read more clearly.

Table 1. Statistical information on collected construction accidents

Accident type

Frequency

Construction project type

Frequency

Lifting equipment collapse

31

Urban infrastructure project

24

Building collapse

29

Plant construction and maintenance project

35

Object falling at high altitude

26

Civil residential project

23

Personnel falling from high altitude

24

National Engineering Construction Project

9

Fire accident

14

Public civil construction project

19

Electric shock accident

11

Commercial building project

25

Statistics of Accident Location

Frequency

Statistics of casualties in construction accidents

Frequency

First-tier cities

21

Minor wound

21

Second-tier cities

31

Serious injury

16

Third-class cities

49

One to two deaths

41

Fourth-tier cities

34

Three to five deaths

32

More than five man died

25

In order to make the sample more representative, we have added construction accident reports from the official website of the Ministry of Housing and Construction of the People’s Republic of China in 2020 and 2019, increasing the total number of construction accident reports from 78 to 135. The criteria for the selected accident report must include the basic situation of the accident unit, the accident occurrence, the accident site investigation and technical analysis, the casualty of the accident, and the direct and indirect cause analysis of the accident, ensure the integrity and referability of subsequent incident analysis.

Secondly, according to your suggestion, we have added qualitative analysis to Nvivo 12. Such as “Nvivo 12 is the software applied to qualitative analysis. The internals folder in Nvivo 12 can contain diverse types of sources, such as documents, PDFs, data sets, audio/video sources, and pictures. The data set often includes structured data that is arranged in records (rows) and fields (columns), which cannot be edited in NVivo. This study developed a data set by importing a spreadsheet into Nvivo 12, which analysis by Open coding, Spindle coding, Selective coding.”

Finally, before we finished the paper, we had invited three experts in the construction industry to verify and evaluate our coding results. They all believed that they were logical and factual, but they were not reflected in the article. Based on your suggestion, we put the description in the text, such as“In order to ensure the authority and representativeness of the research results, three construction industry experts (including two Chinese construction engineers and a university professor) were invited to evaluate the coding results, all of which indicated that the coding results are logical and factual.”

Point 4: Results

Table 1, 2 and 3 should be simplified: main data should be concentrated. Typical construction accidents should comprise a detailed qualitative analysis that allows demonstrating the sufficiency of data to justify the diagram proposed in Figure 6.

Response 4: Thanks for the suggestions about the Results. Based on your suggestions, we added more a detailed qualitative analysis of typical accidents to justify the diagram proposed in Figure 6.

Such as “Identification: If a precursor is not enough to cause the accident and the relevant subjects have not been identified in time, this will not lead to a serious accident. However, if a precursor is the main factor that directly causes the accident and is not recognized, the precursor will finally result in accidents with serious consequences. Such as: P1 precursor is not discovered, the precursor continues to develop into a critical event, but due to insufficient necessary conditions to cause the accident, it does not constitute a serious accident, there is opportunism. P2 is also undiscovered and contains the precursors that mainly caused the accident. The precursors developed, serious accidents occured. Therefore, the identification stage is the beginning of the entire development process of precursors.

Response: When a precursor is identified, whether workers take actions against it or not will directly impact the development and evolution of the precursor. If a precursor is the main cause of accidents and workers do not take any action to deal with it, the precursor event itself will directly lead to accidents with serious consequences. Such as P3: The workers have found that the construction lifts are not equipped with safety gates and are in an unsafe state. However, no measures have been taken to prevent the precursor, and the precursor can cause an accident, leading to a serious accident. The reaction stage is the second stage of the development process of precursor.

Effectiveness testing: The effectiveness of the used action to deal with precursors will impact the performance of precursor management. The construction process is affected by a variety of factors that are related to environment, construction equipment, and construction atmosphere. In addition, workers’ ways of responding to precursors are different. It becomes imperative to test the usefulness of adopted action. If precursors are identified and reasonable and effective risk response measures are taken, construction accidents can be largely avoided. Such as P4 and P5: the worker only reported to the construction team leader and has not been directly reported to the effective influencer, and the adoption of invalid evasive measures has led to serious construction consequences. No effective measures are generated, leading to serious accidents. P5: Worker A found precursor and stopped the construction work B, reported to the superior supervision department, the logistics department in time. There were no serious consequences. Effectiveness testing is the last link in the development of precursors.”

We also think that Table 1 is the output of the whole result after the grounded theoretical analysis, which can fully show the analysis content to readers. Table 2 shows an indicator system of precursors for construction accidents. Table 3 shows the proportions of the main precursor events of different types of accidents. They are the basis for the analysis of the discussion part. We want to keep it to ensure the integrity of the article. Thank you again.

Point 5: Discussion

The discussion does not include limitations, relationships with previous articles and research on the subject, the techniques usually applied in construction and future proposals. An intrinsic review of this part is necessary to ensure that it is not confused with the results section.

Response 5: We really appreciate the reviewer for this comment. Based on your comments, we made the following corrects and comments:

Firstly, we put limitations into the conclusion of the article to elaborate, as a prospect for future work. We hope you understand it very much. Such as “The study has some limitations that should be well considered and improved in future studies. For instance, the study is based on the grounded theory which is more based on qualitative analysis. Future studies can consider the usefulness of quantitative analysis methods to explore the research topic from a different perspective.”

Secondly, we have added relationships with previous articles and research on the subject to make the article more contrastive. Such as “Different from discussing the relationships and differentiations accident, near miss, and unsafe behavior/condition [25], this study analysis of different types of building accidents regarding major precursors.”

Finally, the techniques usually applied in construction was put in the literature review to let the reader first understand the analytical techniques used in this article. And the future proposal was put in conclusion, such as “Future studies can consider the usefulness of quantitative analysis methods to explore the research topic from a different perspective.”

Point 6: Conclusion

The conclusion does not provide an interesting outcome. This would be possible if a thorough discussion of the current subject in construction techniques had been conducted.

Response 6: It is our great honor to receive this valuable comment, which is very helpful for us to better refine our interesting outcome. We re-summarized the contributions and conclusions in the article, and revised the conclusions based on your suggestions. Interesting innovations mainly include the following:

Firstly, this paper uses grounded theory to forms a precursor indicator system, and the core elements of this system which included two major categories of precursors: behavioral factors and physical factors and five minor categories of precursors: individual behavior factors, organizational driving factors, objective physical factors, construction environmental factors, mechanical equipment factors.

Secondly, based on analyzing the process of a precursor evolving into a construction accident, precursor management strategy that includes the three steps of identification, response, and effectiveness testing is proposed to facilitate the safety management on construction sites.

Finally, based on the analysis of typical cases, we found that in the order of organizational-driven precursor, individual behavior precursor, and objective physical precursor, the detectability and recognizability continue to increase, but the internal influence and ambiguity continue to weaken in Fig. 6. Among the two major types of accidents, collapse and fall, individual behavior and organizational drive account for the highest proportion in Fig. 7.

Fig. 6. Diagram of changes in the three types of precursors

Fig. 7. Types of precursors in the high-altitude falling accidents

Point 7: Bibliography

The references do not comply with the standard of IJERPH.

Response 7: Thanks for the suggestions about applying the standard of IJERPH into the references. Based on your suggestion, we have completely revised the reference section to meet the standard of IJERPH.

Such as: Kyriakidis, M., Hirsch, R., Majumdar, A. Metro railway safety: An analysis of accident precursors. Saf. Sci.2012, 50(7), 1535-1548.

Makki, A. A., & Mosly, I. Determinants for Safety Climate Evaluation of Construction Industry Sites in Saudi Arabia. Int. J. Environ. Res. Public Health, 2020,17(21), 8225.

In all, we found the referee’s comments are quite helpful, and we revised our paper point-by-point. We thank you overall for your valuable suggestions that have helped us to craft a stronger manuscript. Thank you for your help.

Yours sincerely,

Qingfeng Meng

Name: Qingfeng Meng

E-mail: mqf@ujs.edu.cn

cc.

Zhen Li; Rui Mao; Xin Hu; Hongxian Li

Reviewer 3 Report

First of all, I would like to congratulate but there are some points that need to be adressed.

  1. The authors need to follow the style of the journal, such as the affiliation or the cite style. Right now, the are some mistakes about the styling of the manuscript.
  2. The authors used data from 2003, is there more updated datan Asia? (from line 25-30). This can also be applied to the literature review, which should include more recent and relevant references. 
  3. When was the data retrieved from the Ministry of Housing and Construction of the People’s Republic of China?
  4. Authors should expelled what us Nvivo 12 since the readers may not know it.
  5. The steps taken to decided the coding should be further explained since readers could not understand the reason beneath it. 
  6. The discussion focused on the results but don't compared with the hypothesis from previous work or other results. 

Author Response

Point 1: The authors need to follow the style of the journal, such as the affiliation or the cite style. Right now, the are some mistakes about the styling of the manuscript.

Response 1: Thanks for the suggestions about the style of the journal. According to your suggestion, we have revised the citation format, affiliation, table and figure styles and language style of the article.

Such as: Kyriakidis, M., Hirsch, R., Majumdar, A. Metro railway safety: An analysis of accident precursors. Saf. Sci.2012, 50(7), 1535-1548.

Makki, A. A., & Mosly, I. Determinants for Safety Climate Evaluation of Construction Industry Sites in Saudi Arabia. Int. J. Environ. Res. Public Health, 2020,17(21), 8225.

And such as:

Table 1. Statistical information on collected construction accidents

Accident type

Frequency

Construction project type

Frequency

Lifting equipment collapse

31

Urban infrastructure project

24

Building collapse

29

Plant construction and maintenance project

35

Object falling at high altitude

26

Civil residential project

23

Personnel falling from high altitude

24

National Engineering Construction Project

9

Fire accident

14

Public civil construction project

19

Electric shock accident

11

Commercial building project

25

Statistics of Accident Location

Frequency

Statistics of casualties in construction accidents

Frequency

First-tier cities

21

Minor wound

21

Second-tier cities

31

Serious injury

16

Third-class cities

49

One to two deaths

41

Fourth-tier cities

34

Three to five deaths

32

More than five man died

25

Point 2: The authors used data from 2003, is there more updated datan Asia? (from line 25-30). This can also be applied to the literature review, which should include more recent and relevant references.

Response 2: It is my great honor to get your suggestions, which is of great importance to let me update the more recent and relevant references. We have updated the more recent China data: “According to the statistics of the Emergency Management Department of the People’s Republic of China in the first half of 2018, there were 1732 accidents and 1752 fatalities in the construction industry.”

  And we have applied to the more recent literature review in this article.Such as: “Recently, safety climate significantly influences safety performance, making research in the field of safety climate a vital step toward raising safety levels at construction sites.” from“Makki, A. A., & Mosly, I. (2020). Determinants for Safety Climate Evaluation of Construction Industry Sites in Saudi Arabia. International Journal of Environmental Research and Public Health, 17(21), 8225.”

Point 3: When was the data retrieved from the Ministry of Housing and Construction of the People’s Republic of China?

Response 3: It is my great honor to get your question. Because we re-reviewed and increased the number of accident cases during the modification phase, the construction accident reports were retrieved and collected in the 2020s. We have indicated the data in the article. Corrected as follows: “The construction accident reports were retrieved and collected from the official website of the Ministry of Housing and Construction of the People’s Republic of China in the 2020s. This can ensure the accuracy and authority of data collected and used in this study. The keywords adopted to search for construction accident reports include “building accident report”, “collapse accident report”, and “falling accident report”. The criteria for the selected accident report must include the basic situation of the accident unit, the accident occurrence, the accident site investigation and technical analysis, the casualty of the accident, and the direct and indirect cause analysis of the accident, ensure the integrity and referability of subsequent incident analysis. Finally, 135 representative accident reports were selected and used in this study. All the used accident cases in this study occurred during 2013-2020.”

Point 4: Authors should expelled what us Nvivo 12 since the readers may not know it.

Response 4: We really appreciate the reviewers for this comment. We have found that we lacked how to use Nvivo 12. We have made in-depth consideration and modification, adding sufficient Nvivo 12 details to let readers know it. Such as:

“The criteria for the selected accident report must include the basic situation of the accident unit, the accident occurrence, the accident site investigation and technical analysis, the casualty of the accident, and the direct and indirect cause analysis of the accident.”

“It organizes the initial sentences with the same or similar meaning, finds out the identifiable phenomenon, and pastes the initial code or label. Then, it gathers labels to form a concept and extracts categories. The retrieved information is condensed into several categories to form a conceptual system. Line-by-line or sentence-by-sentence is the most efficient way to conduct open coding analysis.”

“Spindle coding as a set of procedures whereby data were put back together in new ways after open coding, by making connections between constructs. The purpose of spindle coding is to aggregate the free nodes from the previous step and group them into parent nodes and child nodes. Some similar nodes are merged and renamed. The first level nodes are often called parent nodes. The others are child nodes. A child node has one parent node at most. For example, the open coding notes of “violation of crossing the railing to the crane running area”, “overhauling the worker’s illegal operation” can form the spindle paradigm of the worker's illegal operation behavior.”

“Selective coding is the process of selecting the core constructs. Parent nodes and child nodes from axial coding are merely descriptions of the data which cannot exhibit the multiple relationships among them. It is necessary to further explore the complexity of the multiple relationships and capture the nature of precursors. The objective of selective coding is to develop a single storyline around which everything else is draped.”

Point 5: The steps taken to decided the coding should be further explained since readers could not understand the reason beneath it. 

Response 5: Thank you for your comment and suggestion. According to your suggestion, we have further explained the steps taken to decide the coding. Such as:

“Open coding is the process of decomposing and comparing the original text to conceptualize and categorize the encoding process. It organizes the initial sentences with the same or similar meaning, finds out the identifiable phenomenon, and pastes the initial code or label. Then, it gathers labels to form a concept and extracts categories. The retrieved information is condensed into several categories to form a conceptual system. Line-by-line or sentence-by-sentence is the most efficient way to conduct open coding analysis”

“Selective coding is to identify the core categories and connect different categories, which refers to the process of conceptualizing the untapped and complete scope.”

 “The objective of selective coding is to develop a single storyline around which everything else is draped.  The path diagram of precursor development in construction and the management of accident precursor are put in the discussion for research.”

Point 6: The discussion focused on the results but don't compared with the hypothesis from previous work or other results.

Response 6: Thanks for this comment and the consideration of these problems is very helpful to improve the Logical and Completeness of this paper.

We have made the following rectifications in the discussion based on your suggestions:

Firstly, while proposing the results, we added an analysis of the above data results and used figures to illustrate them. Such as “Individual behavioral and Organization-driven factors are the dominant precursors in the high-altitude falling accidents from Fig. 7, occupying 30% and 33%. The reasons to these accidents include weak safety awareness, lack of safety handling abilities, and some subjective factors that affect behaviors. This kind of precursor is not easy to identify before the occurrence of accidents, and lacks methods of real-time monitoring. In order to avoid accidents caused by such precursory events, one of the suggested strategies is safety training which can improve workers’ safety consciousness, strengthen safety supervision, and form a safety culture and atmosphere.”

Fig. 7. Types of precursors in the high-altitude falling accidents

Secondly, comparing the above-mentioned typical accidents, put forward its own precursor management measures. Such as “P2 and P4 in Table 5, the indirect cause of accidents is related to the organization, whereas the direct cause is usually individuals’ poor performance. Before the occurrence of construction accidents, individual behavioral precursors emerged.”

Finally, we found the characteristics of the same type of articles in the literature review and put forward our own unique viewpoints. Such as “Compared with the traditional precursor management [8],it was divided into three stage: identification, Response, Effectiveness stage.”

In all, we found the referee’s comments are quite helpful, and we revised our paper point-by-point. We thank you overall for your valuable suggestions that have helped us to craft a stronger manuscript. Thank you for your help.

Yours sincerely,

Qingfeng Meng

Name: Qingfeng Meng

E-mail: mqf@ujs.edu.cn

cc.

Zhen Li; Rui Mao; Xin Hu; Hongxian Li

Round 2

Reviewer 1 Report

The work has really developed and the author has met all my previous comments. 

Author Response

Response to Reviewer 1 Comments

Point 1: The work has really developed and the author has met all my previous comments.

Response 1: We really appreciate you for valuable comments on our paper, which has improved the logic and correctness of the paper. We wish you have a happy work and life.

In all, we found the referee’s comments are quite helpful. We thank you overall for your valuable suggestions that have helped us to craft a stronger manuscript. Thank you for your help.

Yours sincerely,

Qingfeng Meng

Name: Qingfeng Meng

E-mail: mqf@ujs.edu.cn

cc.

Zhen Li; Rui Mao; Xin Hu; Hongxian Li

Reviewer 2 Report

LITERATURE REVIEW

The section of “2.1 Construction accidents and safety management” need to be actualized with the necessity of new techniques to analyse ris factors and recent articles.

I give you some tips to introduce reference and the multi-methodological analysis with connect with the precursors of accidents.

In lines 69-71 you when you says “Recently, safety climate significantly 69 influences safety performance, making research in the field of safety climate a vital step toward raising safety levels at construction sites [17]. All these studies have contributed to a better understanding of construction safety management”.

You can modify the sentence including new management techniques applied in construction: Recently, safety climate significantly influences safety performance, making research in the field of safety climate a vital step toward raising safety levels at construction sites [17]. Furthermore, recent authors have indicated the necessity of use multi-methodological analysis under different perspectives, i.e. biomechanical risk analysis (Cite here the following: https://www.revistadyna.com/search/ergonomic-risk-factors-analysis-with-multi-methodological-approach-assessing-workers-activities-in-b) and multi-management or multi-criteria methods, i.e. https://www.tandfonline.com/doi/abs/10.1080/10807039.2018.1424531 or https://core.ac.uk/download/pdf/236052305.pdf, among others). These studies have contributed to a better understanding of construction safety management and the need to utilize nee management practices to risk analysis.

RESEARCH METHODS

3.1 Data collection

The sample indicated (135) has been mentioned as “representative accident reports”, but this consideration is not clarified. Unless you have introduced data with statistical information, these data do not justify the selected sample. I mean, you need to calculate sample size and power analysis (see, i.e. https://www.statisticssolutions.com/sample-size-calculation-and-sample-size-justification/ and you have here an online calculator: https://www.surveysystem.com/sscalc.htm). I suggest considering the global population of accidents occurs during 2013-2020 and provide the general calculation. In addition to this calculation, add that the sample was reduced because some criteria were considered. These criteria could be type of accidents with higher risk such as falling from high altitude, object falling, among others.

DISCUSSION AND CONCLUSION

Data provided in the discussion continues to be mixed with results. The titles of section 5.1., 5.2. and 5.3 comprise the main details of the results. So, this section should be included in the results as final analysis. The following paragraphs should be reserve for the discussion:

Lines 281-288: The physical precursors can be avoided or reduced by behavioral precursors. In the case of physical precursors as the dominant factor, the physical precursors can directly lead to construction accidents, but can be identified and avoided through individual behavior and organizational drivers. It is important to implement real-time identification of pre-cursors during the construction process to facilitate the management of precursor events. Based on the result and analysis of typical cases, it is found that in the order of organizational-driven precursor, individual behavior precursor, and objective physical precursor, the detectability and recognizability continue to increase, but the internal influence and ambiguity continue to weaken.

Lines 304-308: This kind of precursor is not easy to identify before the occurrence of accidents, and lacks methods of real-time monitoring. In order to avoid accidents caused by such precursory events, one of the suggested strategies is safety training which can improve workers’ safety consciousness, strengthen safety supervision, and form a safety culture and atmosphere.

Lines 319-321 (the second is a future line): Construction workers’ insufficient safety knowledge and awareness makes them un-sensitive to precursors. Thus, it is necessary to improve safety awareness and knowledge of workers, and develop effective methods and techniques for identifying precursors.

Lines 322-328: At the Response stage, the response to precursors is affected by such factors as safety awareness, organizational safety atmosphere, subjective attitude, and regulatory pressure. It is suggested to organize and carry out safety responsibility training to improve construction professionals’ sense of responsibility to handle precursors.

(Now the future lines comes): In addition, the inspection system and the supervision system should be established, and the regulations regarding the reward and punishment in the process of handling precursors should be well developed.

Lines 329-333(future lines): At the Effectiveness testing stage, it is important to ensure that effective responses are proposed given that using effective response. However, a lot of uncontrollable factors impact the effectiveness of the strategies used such as the nature of precursors and the abilities of managers. It is important that all project stakeholders can work together to understand these factors and handle precursors in timely, effective and reasonable ways.

A quality discussion should be written, preferably without subsections.

So, they need to provide the following parts: Main results obtained, general results with some examples analysed; Limitations; Future lines (all the parts should be included in separated paragraphs). All of this part could be connected with previous researches/papers, i.e. in the sense of management and/or multi-methodological techniques if is necessary (you can cited some articles if you want). Usually, graphics and tables are not included in this section, unless some papers included (see, i.e. the article of Gul).

Conclusion only refers to main “concluded parts of the article”, so, limitations are not included here when you have a section of discussion (see, i.e. the paper of Turskis et al.). Limitations should be commented in discussion and, if you want, referred basically in discussion (see, i.e. the paper of Zorrilla-Muñoz et al.).

Other proposal is to mention “Results and discussion” and develop the chapter “Limitations” and “Conclusion”(see the paper of Sadeghi et al.).

Author Response

Response to Reviewer 2 Comments

Point 1: LITERATURE REVIEW

The section of “2.1 Construction accidents and safety management” need to be actualized with the necessity of new techniques to analyse ris factors and recent articles.

Response 1: It is my great honor to get your suggestions. We downloaded new technologies and articles about risk factor analysis for reading. Based on your comments, the relevant content in the literature review was revised and related articles were cited to make the literature review more logical. The specific modifications are as follows:

“Recently, safety climate significantly influences safety performance, making research in the field of safety climate a vital step toward raising safety levels at construction sites [17]. Furthermore, recent authors have indicated the necessity of use mul-ti-methodological, multi-management and multi-criteria methods analysis under dif-ferent perspectives [18-19], such as biomechanical risk analysis [20]. These studies have contributed to a better understanding of construction safety management and have utilized management practices to risk analysis.”

Point 2: RESEARCH METHODS

3.1 Data collection

The sample indicated (135) has been mentioned as “representative accident reports”, but this consideration is not clarified. Unless you have introduced data with statistical information, these data do not justify the selected sample. I mean, you need to calculate sample size and power analysis. I suggest considering the global population of accidents occurs during 2013-2020 and provide the general calculation. In addition to this calculation, add that the sample was reduced because some criteria were considered. These criteria could be type of accidents with higher risk such as falling from high altitude, object falling, among others.

Response 2: Thanks for this comment and the consideration of these problems is very helpful to improve the accuracy and logic of the sample.

Based on your suggestions, we made the following modifications and explanations: This article is based on the research on China's construction safety accidents, and through the construction safety accident statistics report published on the official website of the Ministry of Construction of China, there are about 8264 construction accident reports in 2013-2019, including the incomplete statistics of the 2020 construction industry safety accident reports. There are about 9000 cases. According to the calculation method of sample selection, a 95% confidence level and a confidence interval of 8 can be used to calculate a reasonable sample size of 148. Therefore, this paper selects 135 construction accident reports within a reasonable range.

Point 3: DISCUSSION AND CONCLUSION

Data provided in the discussion continues to be mixed with results. The titles of section 5.1., 5.2. and 5.3 comprise the main details of the results. So, this section should be included in the results as final analysis.

A quality discussion should be written, preferably without subsections.

So, they need to provide the following parts: Main results obtained, general results with some examples analysed; Limitations; Future lines (all the parts should be included in separated paragraphs). All of this part could be connected with previous researches/papers, i.e. in the sense of management and/or multi-methodological techniques if is necessary (you can cited some articles if you want). Usually, graphics and tables are not included in this section, unless some papers included (see, i.e. the article of Gul).

Conclusion only refers to main “concluded parts of the article”, so, limitations are not included here when you have a section of discussion (see, i.e. the paper of Turskis et al.). Limitations should be commented in discussion and, if you want, referred basically in discussion (see, i.e. the paper of Zorrilla-Muñoz et al.).

Other proposal is to mention “Results and discussion” and develop the chapter “Limitations” and “Conclusion”(see the paper of Sadeghi et al.).

Response 3: It is my great honor to get your suggestions, which is of great importance to improve the readability and logic of the discussion and conclusion. According to your suggestion, we made the following improvements:

First, in order not to mix the discussion with the results, we cancelled the title of section 5.1., 5.2. and 5.3, so that this part is no longer discussed in three sections.

Second, citing relevant articles to compare with the identification path and research results of this paper, making the article more logical and comparable. At the same time, I hope that the discussion part is based on the previous research and keeps the readability, and keeps the relevant result pictures. Hope the reviewer can understand.

Third, we separate the limitations of this article and the points that can be developed in the future to form a new paragraph. Allow readers to conduct quantitative in-depth research on the basis of this article.

Finally, we revised the content and structure of discussions, restrictions, and conclusions based on the reviewers’ comments. In the discussion, the content of the future development direction was modified to make the article more logical and developmental.

In all, we found the referee’s comments are quite helpful, and we revised our paper point-by-point. We thank you overall for your valuable suggestions that have helped us to craft a stronger manuscript. Thank you for your help.

Yours sincerely,

Qingfeng Meng

Name: Qingfeng Meng

E-mail: mqf@ujs.edu.cn

cc.

Zhen Li; Rui Mao; Xin Hu; Hongxian Li

Reviewer 3 Report

The authors have greatly improved the manuscript.

But, I would recommend that the authors describe more the tables, especially, table 3 and 4.

Other thing is the format which is odd or rare.Has this happened because of the pdf?

Author Response

Response to Reviewer 3 Comments

Point 1: I would recommend that the authors describe more the tables, especially, table 3 and 4.

Response 1: Thanks for the suggestions about describing more the tables, especially, table 3 and 4. According to your suggestion, we have added the chart formation method and process, and the proportion of each type of chart. At the same time, the qualitative description of the table was revised.

Such as: Table 3 shows the indicator system of the precursor influencing factors for con-struction projects. By counting the frequency of various precursor events occurrence in the open coding process, it can be determined the proportions of Individual behavioral factors (36.2%), Organizational drivers (29.2%), Objective physical factors (15.6%), Construction environmental factors (8%) and Mechanical Equipment factors (10.8%) in the indicator system. As shown in Table 3, the factors can be grouped into the two categories of Behavioral factors and Physical factors, and the indicator system is dom-inated by behavioral factors and supplemented by physical factors. The two categories can be further grouped into several sub-categories which can be explained by open coding results. It also shows that Personal safety awareness is the most important im-pacting factor of accident precursors in the construction industry.

Such as: By counting the frequency of various precursors occurrence in each type of acci-dent, it can be found that different types of accidents have different leading precursors. As shown in Table 4, the collapse and high drop accidents accounted for the highest proportion of construction accidents, 44.4% and 37% separately. Lack of safety su-per-vision and Weak awareness of safety are identified as the most important precur-sors that result in collapse accidents, 29 and 32 times separately. Whereas Imperfect implementation of safety management system is most important for falling from a height accident. Weak awareness of safety and Lack of inspection of infrastructure are identified as two key precursors for fire accidents,16 and 15 times separately. Com-pared with electric shock accidents, Failure to follow the prescribed operating proce-dures is the most important.

Point 2: Other thing is the format which is odd or rare. Has this happened because of the pdf?

Response 2: We are so sorry to make you feel that there is a problem with the format. This template is downloaded from the official website of the journal, and the text table and pictures may have some matching problems. We have tried our best to revise and hope the editor will revise the publication format.

In all, we found the referee’s comments are quite helpful. We thank you overall for your valuable suggestions that have helped us to craft a stronger manuscript. Thank you for your help.

Yours sincerely,

Qingfeng Meng

Name: Qingfeng Meng

E-mail: mqf@ujs.edu.cn

cc.

Zhen Li; Rui Mao; Xin Hu; Hongxian Li